# From *Epimedium* to Neuroprotection: Exploring the Potential of Wushanicaritin

**DOI:** 10.3390/foods13101493

**Published:** 2024-05-11

**Authors:** Donghui Luo, Dingding Shi, Lingrong Wen

**Affiliations:** 1Chaozhou Branch of Chemistry and Chemical Engineering Guangdong Laboratory, Chaozhou 521000, China; luodonghui@bio-th.com; 2Guangdong Provincial Key Laboratory of Applied Botany, South China Botanical Garden, Chinese Academy of Sciences, Guangzhou 510650, China; sdingding17@163.com; 3University of Chinese Academy of Sciences, Beijing 100049, China

**Keywords:** *Epimedium*, intercellular antioxidant activity, neuroprotective activity, oxidative stress, antiapoptotic effect

## Abstract

*Epimedium* has been used for functional foods with many beneficial functions to human health. Wushanicaritin is one of the most important chemicals int *Epimedium*. This study investigated the neuroprotective effects of wushanicaritin and potential underlying mechanisms. The results demonstrated that wushanicaritin possessed superior intercellular antioxidant activity compared to icaritin. Wushanicaritin, with an EC_50_ value of 3.87 μM, showed better neuroprotective effect than quercetin, a promising neuroprotection agent. Wushanicaritin significantly reversed lactate dehydrogenase release, reactive oxygen species generation, cell apoptosis, and mRNA expression related to cell apoptosis and oxidative defense, in glutamate-induced PC-12 cells. Wushanicaritin could also maintain the enzymatic antioxidant defense system and mitochondrial function. The suppression of caspase-3 activation and amelioration of mitochondrial membrane potential loss and nucleus morphology changes were involved in the antiapoptotic effect of wushanicaritin. These findings suggested that wushanicaritin possesses excellent intercellular antioxidant and neuroprotective activities, showing potential promise in functional foods.

## 1. Introduction

Neurological diseases, such as Alzheimer’s disease, multiple sclerosis, and Parkinson’s disease, have garnered increasing attention because of the rising prevalence of aging populations [1]. Among them, Alzheimer’s disease is recognized as the primary cause of cognitive impairment and dementia in aging individuals, resulting in serious humanistic and economic burdens, and has become a worldwide problem. However, current effective prevention and treatment methods for these diseases remain elusive. Consequently, there is a pressing need for innovative therapeutic strategies and agents to prevent neurological disorders [2,3]. Despite the ambiguity surrounding the etiology of neurodegenerative diseases such as Alzheimer’s disease, numerous reports have shown that an overabundance of reactive oxygen species (ROS) is closely associated with neuronal damage in various neurodegenerative disorders, including brain trauma, Alzheimer’s disease, cerebral ischemia, and Parkinson’s disease [3]. Generally, limited quantities of ROS are generated within cells and play a crucial role in maintaining essential cellular functions and homeostasis [4]. However, oxidative stress can be induced and triggered because of an imbalance between ROS generation and obliteration. This stress can cause necrotic and apoptotic processes in cells, finally resulting in irreversible damage and/or cell death [4,5]. The brain, being the primary site of oxygen metabolism, is particularly susceptible to oxidative stress. An overproduction of ROS in the brain can result in detrimental effects, particularly neuronal apoptosis or necrosis [6]. Therefore, oxidative stress plays a significant role in the progression of neurological diseases, necessitating the use of antioxidants for their prevention and/or treatment [7]. Neuroprotective strategies for ROS-mediated cellular oxidative stress have proven effective against neurodegenerative diseases [8,9].

*Epimedium* (*Berberidaceae*), commonly referred to as barrenwort, is globally utilized for both food and medicinal purposes [10]. Additionally, the leaves, stem and rhizome can serve as a therapeutic remedy. Pharmacological studies have demonstrated that *Epimedium* has significant regulatory effects on bone development and the human immune system. Moreover, it has been shown to enhance cardiovascular function and exhibits anti-aging, anti-cancer, anti-osteoporosis, and neuroprotective properties [11]. The chemicals in *Epimedium* exhibited significant potential for utilization as nutraceuticals in functional foods or as phytopharmaceuticals in the prevention and treatment of chronic diseases [12]. Although over 260 compounds have been identified in *Epimedium*, the major bioactive chemicals responsible for its pharmacological activities, such as anti-aging and neuroprotection, remain largely unexplored. Prenylated flavonoids have been reported to be the characteristic bioactive constituents in *Epimedium* [13]. Wushanicaritin, one of the most important prenylated flavonoids in *Epimedium*, exhibited significant DPPH radical scavenging activity comparable to vitamin C [14]. Additionally, wushanicaritin showed anti-inflammatory effects in vitro and in vivo [15]. These results suggest that wushanicaritin may exhibit significant neuroprotective properties. However, to date, there has been limited information available on this subject.

In this study, we assessed the intercellular antioxidant activity of wushanicaritin. Furthermore, the neuroprotective effects of wushanicaritin against glutamate-induced neurotoxicity in PC-12 cells, a well-established neuronal model, were also evaluated. The underlying mechanisms against oxidative stress and apoptosis were investigated. These findings provide insights into the potential of wushanicaritin as a neuroprotective agent.

## 2. Materials and Methods

### 2.1. Chemical Regents

Glutamate, 2, 2′-azobis (2-methylpropionamidine) dihydrochloride (AAPH), dimethyl sulfoxide (DMSO), and 2′,7′-dichlorodihydrofluorescein diacetate (DCFH-DA) were provided by Merck Limited (Shanghai, China). Quercetin (97%, HPLC) was obtained from Shanghai Macklin Biochemical Technology Co., Ltd. (Shanghai, China). Phosphate-buffered saline (PBS), RPMI 1640 medium, William’s medium E, Hank’s balanced salt solution (HBSS), 4-(2-hydroxyethyl)-1-piperazineethanesulfonic acid (Hepes), and fetal bovine serum were obtained from Thermo Fisher Scientific (China) Co., Ltd. (Shanghai, China). Assay kits, including Hoechst 33342 (100×) staining, Cell counting kit-8 (CCK8), lactate dehydrogenase (LDH) release, catalase (CAT) activity, Bicinchoninic acid (BCA) assay, Bradford Protein Assay, Caspase 3 Activity Assay, and ATP Assay were purchased from Beyotime Biotechnology (Shanghai, China). Mitochondrial Membrane Potential Assay Kit with 5,5′,6,6-tetrachloro-1,10,3,30-tetraethylbenzimidazol-carbocyanine iodide (JC-1) was provided by Solarbio Life Sciences (Beijing, China). Other kits for the determination of superoxide dismutase (SOD) and glutathione peroxidase (GSH-Px) activities were purchased from Nanjing Jiancheng Bioengineering Institute (Nanjing, China). Icaritin and wushanicaritin were prepared previously [16]. HiPure total RNA Mini kit was provided by Magen Biological Technology Co., Ltd., (Guangzhou, China).

### 2.2. Cell Culture

HepG_2_ and PC-12 cells (highly differentiated) were kindly provided by the Cell Bank and Stem Cell Bank of the Chinese Academy of Sciences (Shanghai, China), respectively. HepG_2_ cells were cultured in a growth medium (William’s medium E) that contained 5 μg/mL insulin, 10 mM Hepes, 2 mM L-glutamine, 100 μg/mL gentamicin, 50 units/mL penicillin, 0.05 μg/mL hydrocortisone, 50 μg/mL streptomycin, and 5% fetal bovine. The PC-12 cells used in this study were between passages 5 and 10 and were cultured in growth medium (RPMI1640 medium) supplemented with Hepes (10 mM), fetal bovine serum (10%), streptomycin, penicillin and L-glutamine (2 mM). All cells were grown at 37 ºC in a cell incubator with 5% CO_2_. 

### 2.3. Preparation of Solutions

The stock solutions of quercetin, icaritin and wushanicaritin were initially prepared in DMSO to a concentration of 100 mM. Their working solutions were subsequently diluted in the growth medium with or without serum treatment. A stock solution of 200 mM AAPH was formulated in H_2_O. Prior to use, the stock solution was further diluted in an oxidant treatment medium (phenol-free HBSS supplemented with 10 mM Hepes). A stock solution of 200 mM DCFH-DA was formulated in methanol. Prior to use, the DCFH-DA solution was further diluted in serum-free growth medium to yield a working solution of 50 or 10 μM, respectively. 

### 2.4. Cellular Antioxidant Activity Assay

The assay for cellular antioxidant activity (CAA) was performed as previously described [17]. Following a 24 h incubation period, HepG2 cells were treated with antioxidant medium, which was William’s medium E containing 10 mM Hepes, 2 mM L-glutamine, 50 μM DCFH-DA and various concentrations of quercetin (2–20 μM), icaritin (2–40 μM) or wushanicaritin (2–40 μM). After a one-hour incubation, the antioxidant medium was entirely removed, and the cells underwent either PBS wash or no PBS wash treatments. Following triplicate treatments with AAPH, a 96-well microplate was positioned into a multi-mode microplate reader (Spark, Tecan Group Ltd., Männedorf, Switzerland) at 37 °C, utilizing an emission wavelength of 538 nm and an excitation wavelength of 485 nm. Fluorescence generation was continuously monitored every 5 min until 1 h had elapsed. The half maximal effective concentration (EC_50_) and CAA values were calculated as previously described [18]. The concentrations of quercetin, icaritin and wushanicaritin were ascertained through preliminary experiments. They showed no obvious cytotoxicity to HepG2 cells at a treatment time of 12 h.

### 2.5. Cell Viability Determination

A CCK8 assay kit was employed for cell viability determination as reported previously with minor modifications [19]. After a 24 h incubation, PC-12 cells were exposed to growth medium supplemented with varying doses of quercetin (0.1–10.0 μM) or wushanicaritin (0.1–6.0 μM). Subsequently, the cell viability was ascertained via the CCK8 assay after incubating for an additional 24 h. Vehicle control refers to vehicle medium-treated cells. 

### 2.6. Cell Neurotoxicity Induction Assay

To assess the neuroprotective effect of wushanicaritin, we induced damage in PC-12 cells through glutamate exposure, following a previously described protocol with minor modifications [20]. Following a 24 h incubation period, cells were treated with fresh growth medium supplemented with varying doses of wushanicaritin (ranging from 0.05 to 5.00 μM). Each treatment was replicated three times over a 24 h period. Cell viability was subsequently assessed using the CKK8 assay. Vehicle control refers to vehicle medium-treated cells. Quercetin-treated cells were employed as a positive control. The concentrations of quercetin and wushanicaritin were determined based on the results of cell viability assays, and concentration ranges with no obvious cytotoxicity to PC-12 cells were chosen for further study.

### 2.7. LDH and ATP Release Analyses

After glutamate incubation, the LDH activity (U/L) and ATP level (nmol/mg protein) in the culture supernatant were determined by LDH assay and ATP assay kits, respectively, following the guidelines of the manufacturer. Cells with vehicle medium (0.1% DMSO) served as the vehicle control.

### 2.8. Determination of Reactive Oxygen Species (ROS)

The generation of ROS within PC-12 cells was quantified using DCFH-DA staining, following a previously described procedure [19]. The final results were quantified as a percentage, following normalization by the vehicle control.

### 2.9. Hoechst 33342 Staining

After glutamate treatment, Hoechst 33342 (100×) was incorporated into cell culture at 37 °C. Following incubation for 10 min, the cells were meticulously rinsed with cold PBS. After that, a fluorescence microscope (Olympus, Tokyo, Japan) was applied for observation at a magnification factor of 200×.

### 2.10. Mitochondrial Membrane Potential (Δψm) Assay

As reported previously [20], Δψm was ascertained using JC-1 as a fluorescent marker. After staining, the cells were observed under a fluorescence microscope (Olympus, Tokyo, Japan) at a magnification of 100×. 

### 2.11. Cellular Antioxidant Enzymes Activity Determination

PC-12 cells were harvested post glutamate incubation and subsequently rinsed with ice-cold PBS. The resultant cell pellet was then rehydrated in Cell lysis buffer for Western and IP (P0013, Beyotime Biotechnology, Shanghai, China). The CAT, SOD, and GSH-Px activities of the obtained cell lysate were quantified using the respective assay kits. The results obtained were expressed as U/mg protein, and the protein content was determined using the BCA assay kit.

### 2.12. Quantitative Real-Time PCR Assay

As delineated in prior research [20], after incubation with glutamate for 12 or 16 h, total RNA from PC-12 cells was extracted using HiPure total RNA Mini kit, following glutamate treatment. Subsequently, first-strand cDNA synthesis and qRT-PCR analysis were conducted using the PrimeScriptTM RT reagent Kit containing gDNA Eraser and TB Green^®^ Premix TaqTM, respectively, provided by Takara Biomedical Technology Co., Ltd. (Beijing, China). Appendix A shows the PCR primers used in this study. The results were evaluated using the ΔΔCT method, with expression levels normalized against *GADPH* levels within the same sample.

### 2.13. Caspase-3 Activity Determination

Caspase-3 activity was determined as previously described, with minor modification [21]. Briefly, after glutamate incubation for 24 h, all of the cells were collected and lysed using lysis buffer. The caspase-3 activity and protein content of the resultant lysate were determined using Caspase 3 Activity Assay kit and Bradford Protein Assay kit, respectively. The final results were quantified as a percentage, following normalization by the vehicle control, calculating based on the caspase-3 activity per unit weight of protein. 

### 2.14. Flow Cytometry Assay

After glutamate treatment, all of the cells were harvested following centrifugation at a rate of 1000 rpm/min for 5 min. The cells were further subjected to flow cytometry assay to evaluate cell apoptosis using Annexin V–FITC/PI staining, as delineated in prior study [18]. Cells without Annexin V–FITC/PI staining were designated as blanks.

### 2.15. Statistical Analysis

All data were presented as mean ± standard deviation. Statistical analysis was conducted using a one-way analysis of variance (ANOVA). A significance level of *p* < 0.05 or *p* < 0.001 was determined through the use of *t*-test or Dunnett’s T3 test, respectively.

## 3. Results

### 3.1. The Intercellular Antioxidant Activity of Wushanicaritin

CAA assay was employed to evaluate the intercellular antioxidant activities of icaritin and wushanicaritin. The lower cellular fluorescence intensity directly correlates to the prevention of DCFH oxidation, indicating the antioxidant activity against peroxyl radicals. Figure 1 illustrates the kinetics of DCF formation within cells, triggered by peroxyl radicals. The findings suggest that the DCFH oxidation induced by peroxyl radicals was effectively prevented by both quercetin (Figure 1A,B) and wushanicaritin (Figure 1E,F), but not icaritin (Figure 1C,D). This inhibition was observed regardless of whether PBS wash or no PBS wash protocols were used. The dose-dependent nature of this inhibition is evident from the plotted curves. The EC_50_ values of wushanicaritin and quercetin against peroxyl radicals were 15.86 ± 0.83 and 6.99 ± 0.65 µM, respectively, in the PBS wash protocol, whereas in the protocol devoid of PBS wash, these values changed to 16.51 ± 2.11 and 7.23 ± 0.89 µM, respectively. But icaritin demonstrated extremely minimal activity within the examined dose range (2–40 μM), rendering its EC_50_ values undeterminable. The CAA values of wushanicaritin were calculated to be 44.07 ± 2.25 μmol of quercetin equivalents (QE)/100 μmol in No PBS wash protocol, and 43.79 ± 5.55 μmol of QE/100 μmol in PBS wash protocol. All of these results suggested that wushanicaritin was a potential intercellular antioxidant agent.

The CAA assay was devised in response to a demand for a more biologically relevant protocol. This method reflects cellular physiological conditions while also addressing issues of uptake, bioavailability, distribution, and metabolism. Previous study demonstrated that the extracellular antioxidant activity (against DPPH radicals) of icaritin was comparable to that of wushanicaritin [14]. However, in this study, wushanicaritin demonstrated significantly greater intracellular antioxidant activity than icaritin. The structural difference between different flavonoids might influence their uptake, bioavailability, and bioactivity [22,23]. In the case of icaritin and wushanicaritin, the structural difference is the substituent moiety of prenylation. The presence of the 3-OH-3-methylbutyl in wushanicaritin improved intercellular antioxidant activity compared to the 3,3-dimethylallyl in icaritin. Previous studies have shown that the substituent moiety and position of prenylation in flavonoids played a critical role in preventing damage from oxidative stress [24], suggesting that the 3-OH-3-methylbutyl at C-8 was a crucial factor for the intercellular antioxidant activity of prenylated flavonoids. The cellular uptake, quantified as the percentage of CAA values with and without PBS wash, stood at 99.36% for wushanicaritin, suggesting that wushanicaritin was an effective compound that could be internalized by HepG_2_ cells where they react with ROS intercellularly, similarly to quercetin as reported previously [25]. Previous studies also have suggested that incorporating a prenyl moiety into flavonoids can augment their lipophilicity and affinity for p-glycoprotein. This modification enhances the uptake and affinity to cellular membranes, subsequently boosting the bioactivities and bioavailability of the parent flavonoids [26,27]. The higher cellular uptake of wushanicaritin might be attributable to the presence of the 3-OH-3-methylbutyl. 

### 3.2. The Neuroprotective Activity of Wushanicaritin

Our present results indicated that wushanicaritin exhibited potential as an antioxidant agent, demonstrating superior intercellular antioxidant activity. This implies a significant potential of wushanicaritin for neuroprotection. Consequently, wushanicaritin was selected for further study. Prior to analyzing the neuroprotective effect of wushanicaritin, its cytotoxicity against PC-12 cells was elucidated. As depicted in Figure 2A, wushanicaritin possessed no significant cytotoxicity towards PC-12 cells across the tested dose range (0.05~5.00 μM), whereas a significant (*p* < 0.05) reduction in cell viability was observed at a concentration of 6 μM. Thus, a concentration range of wushanicaritin (0.05~5.00 μM) was used for further investigation. Similarly, quercetin showed no cytotoxicity towards PC-12 cells within the tested dose range from 5 to 40 μM, and a dose range of 5 to 40 μM of quercetin was chosen for the neuroprotection assay.

Glutamate incubation resulted in a significant (*p* < 0.05) reduction in cell viability compared to the control group (Figure 2C,D). Notably, when the concentration of wushanicaritin exceeded 1 μM, it significantly (*p* < 0.05) enhanced the cell viability. The results indicated that wushanicaritin could effectively mitigate glutamate-induced PC-12 cell damage within a concentration range of 2–5 μM. The EC_50_ value of wushanicaritin was found to be 3.87 μM, significantly lower than that of quercetin (25.46 μM). This result suggested that despite its lower intercellular antioxidant activity, wushanicaritin provided better neuroprotection compared to quercetin. Additionally, concentrations of wushanicaritin at 2 and 5 μM, which possessed protective rates of 38.31 and 58.07% against glutamate-induced damage, respectively, were selected for further investigation. Quercetin at a concentration of 30 μM was used for positive control.

To further confirm the protective effect of wushanicaritin against glutamate-induced neurotoxicity, the effect of tested compounds on LDH release were studied as well. As displayed in Figure 2E, there was a significant increase in the LDH content after glutamate treatment (41.98 U/L) compared to the vehicle control (20.77 U/L). However, co-treatment with wushanicaritin at 2 μM and 5 μM resulted in a 43.38 and 61.81% reduction in the LDH release induced by glutamate incubation, respectively. However, a higher reduction of 84.96% was observed after co-treatment with quercetin. These results suggested that wushanicaritin treatment could mitigate the LDH release from PC-12 cells induced by glutamate, in a dose-dependent manner. Following cellular membrane damage, LDH is released into culture medium from compromised cell membranes [28]. Therefore, monitored LDH release could serve as an indicator for assessing central neuronal cell injury induced by glutamate. Consequently, it is evident that treatments with wushanicaritin could promote protection against cell membrane damage caused by glutamate.

In recent years, natural products, especially those with better antioxidant activities, have been recognized as viable strategies for the prevention of neurodegenerative diseases and mitigation of associated side effects [9,29]. Furthermore, some bioactive compounds, notably those found in nutritional supplements and functional foods, have garnered significant attention [30]. *Epimedium* has been widely utilized in Oriental countries, serving as a functional food, dietary supplement, or incorporated into wines, teas, tablets, and extracts [31]. As one of the important bioactive compounds found in *Epimedium*, wushanicaritin showed promising intercellular antioxidant activities, suggesting its potential application in functional foods or food supplements for neuroprotection. Moreover, prenylated flavonoids have been proved to be promising nutraceuticals with neuroprotective activity [32,33]. Santi et al. discovered that prenylation in flavonoids enhanced the neuroprotective effect of the parent flavonoid. Moreover, they found that prenylation at C-8 of the A-ring, when accompanied by an unsubstituted B-ring, contributed to a more potent effect [33]. Previous study also suggested that prenylated flavonoids might interact with estrogen receptor α (ERα) to exert their neuroprotective activities [20]. Molecular docking study indicated that wushanicaritin possessed high selectivity towards ERα [16]. All of these results suggested that wushanicaritin might exert its neuroprotective effect through ERα interaction.

### 3.3. The Effects of Wushanicaritin on Oxidative Stress

#### 3.3.1. ROS Production and Activities of Antioxidant Enzymes

Oxidative stress has been identified as a crucial factor involved in the pathogenesis and development of numerous age-related neurodegenerative disorders [8]. This study sought to elucidate the neuroprotective activities of wushanicaritin, which might target oxidative stress, by examining their effects on ROS production and activities of antioxidant enzymes. The fluorescence value in the model group (cells only treated with glutamate) exhibited a significant increase compared to the vehicle control at 234.7%, indicating a significant enhancement for ROS production (Figure 3A). However, when co-culturing with 2 μM and 5 μM wushanicaritin and glutamate, the fluorescence levels decreased to be 162.7% and 134.0%, respectively. In contrast, the fluorescence level following quercetin treatment were recorded at 146.9%. In contrast, glutamate exposure resulted in a reduction in the activities of SOD, CAT and GSH-Px in PC-12 cells (Figure 3B). When compared to the model group, the SOD, CAT and GSH-Px activities in cells from both the wushanicaritin- and quercetin-treated groups were significantly (*p* < 0.05) increased. Especially for wushanicaritin, a concentration-dependent pattern was observed. Generally, the cellular endogenous antioxidant system comprises SOD, CAT and GSH-Px, which function in a synergistic manner to effectively scavenge excessive ROS, thereby mitigating damage [19]. These results suggested that wushanicaritin could effectively facilitate antioxidant enzyme activity to alleviate ROS generation induced by glutamate treatment, similarly to quercetin, a promising agent for Alzheimer’s disease treatment [29].

#### 3.3.2. Transcriptional Expression Levels of Antioxidant Defense-Related Genes

Previous study revealed that transcriptional regulation of associated genes, induced by neuroprotectant, played a crucial role in the prevention of cell damage [34]. To further investigate the mechanisms underlying the neuroprotective effect of wushanicaritin, we examined its impact on the transcriptional levels of genes associated with antioxidant defense. The results showed that glutamate treatment could cause a significant (*p* < 0.05) reduction in the transcriptional levels of *SOD2*, *glutathione peroxidase 1* (*GPx1*), and *CAT*, but only slight reduction in the expression of *SOD1*. However, when compared to the model group, co-treatment with wushanicaritin contributed to a significant up-regulation of the mRNA expression of *SOD1*, *SOD2*, *GPx1*, and *CAT*, in a dose-dependent manner. It is worth noting that the transcriptional level of *SOD2* induced by wushanicaritin was significantly (*p* < 0.05) higher than that of the vehicle control group, suggesting that wushanicaritin might target SOD2 to exert its intercellular antioxidant activity. SOD2, the primary mitochondrial antioxidant enzyme, has been reported to facilitate two molecular transformations of superoxide anion into H_2_O and H_2_O_2_. Overexpression of SOD2 has been shown to improve mitochondrial function and protect cells from glutamate-induced oxidative stress [35]. SOD1 is predominantly localized in the cytoplasm, but a small part of SOD1 is also found in mitochondria. And, mitochondrial SOD1 retains significant physiological relevance by regulating the ROS level in mitochondria below the critical threshold [36]. GPx1, or cellular GPx, is ubiquitously expressed and has the capacity to reduce hydrogen peroxide and fatty acid hydroperoxides, but not esterified hydroperoxide lipids. Previous study indicated that oleuropein could protect cells from oxidative stress by promoting SOD1, CAT and GPx1 expression [37]. Our previous studies also showed that prenylated flavonoids might target activating GPx4 to exert their neuroprotective effects [20,24], suggesting that GPx4 activation might be involved in the neuroprotective of wushanicaritin.

### 3.4. The Effects of Wushanicaritin on Mitochondrial Dysfunction

Mitochondrial dysfunction is recognized as a primary instigator of cellular damage, particularly within brain tissue characterized by high mitochondrial oxygen consumption. The change in Δψm within cells is an important indicator of mitochondrial function. JC-1, a cationic dye, was employed to track the fluctuations in Δψm. In normal cells, JC-1 staining showed red fluorescence under fluorescence microscopy as Δψm is typically high, allowing JC-1 to accumulate within the mitochondrial matrix. Conversely, a depolarization of Δψm can result in the non-accumulation of JC-1 in the mitochondrial matrix, causing a shift in fluorescence from red to green [38]. This trend is illustrated in Figure 4A, which signifies an augmented depolarization of Δψm triggered by glutamate in PC-12 cells. However, a decrease in green fluorescence was found in wushanicaritin-treated cells, in comparison with the model group. Wushanicaritin treatment at increasing concentrations gradually increased cellular red fluorescence, suggested the depolarization of Δψm was improved by wushanicaritin in a dose-dependent manner.

Cellular ATP is predominantly produced by mitochondria via the oxidative phosphorylation mechanism, facilitated by the electron transport chain, which is situated within the internal mitochondrial membrane [4]. As shown in Figure 4B, glutamate treatment resulted in a significant (*p* < 0.05) reduction in ATP synthase, which was only 26.38% of the vehicle control. However, treatment with wushanicaritin and quercetin could significantly reverse the inhibition in ATP synthase induced by glutamate in PC-12 cells. ATP, as the most important energy molecule, plays a pivotal role in various physiological and pathological cellular processes. Alterations in ATP levels can significantly influence cell function. A decrease in ATP levels generally suggests mitochondrial dysfunction. And, regulation of mitochondrial ATP synthase has been considered to be a potential strategy for mitigating mitochondrial dysfunction in Alzheimer’s disease [39]. The enhancement of ATP synthase in wushanicaritin-treated cells implied the repair of mitochondrial function. Mitochondria play a pivotal role in age-related neurodegenerative disorders. Therapeutic interventions that focus on fundamental mitochondrial processes, such as free-radical production and/or energy metabolism, exhibit significant potential for improvement [6]. The present results indicated that wushanicaritin might target improving mitochondrial function to exert a neuroprotective effect.

### 3.5. The Effects of Wushanicaritin on Cell Apoptosis

A reduction in Δψm is a characteristic event observed during the initial phase of cellular apoptosis [21]. Additionally, ATP levels are typically reduced during apoptosis. These results indicated that apoptosis was involved in the glutamate-caused cell death. Therefore, the effect of wushanicaritin on cell apoptosis was also investigated. The primary morphological hallmarks of apoptosis encompass nuclear chromatin condensation, DNA fragmentation, cellular shrinkage, the emergence of apoptotic bodies, and so on. As depicted in Appendix A, following the application of Hoechst 33342 staining, nuclei of normal cells appeared dark blue. However, after a 24 h treatment period with glutamate, nuclei in the model group exhibited either dense intense staining or fragmented blue fluorescence. However, the application of wushanicaritin (2 μM, 5 μM) mitigated the extent of morphological damage observed in PC-12 cells. Furthermore, an increase in the concentration of wushanicaritin amplified this protective effect. The morphological changes in nuclear chromatin serve as a crucial indicator of apoptosis. Therefore, these findings suggest that wushanicaritin might exert neuroprotective effects by mitigating glutamate-induced apoptosis in PC-12 cells.

To confirm the protective effect against cell apoptosis, flow cytometry was performed. As shown by the results in Figure 5, after 24 h of glutamate exposure, the model group exhibited a cell apoptosis rate of 52.3%, markedly higher than that of the vehicle control (3.65%). These cells predominantly underwent late apoptosis. However, following co-treatment with 2 and 5 μM wushanicaritin, the cell apoptosis rates were recorded at 34.63% and 17.51%, respectively, indicating a significant reduction compared to the model group. As the concentrations of wushanicaritin increased, there was a notable decline in the number of apoptotic cells, especially those in the late apoptosis stage. In contrast, the quercetin-treated group demonstrated a pronounced cell apoptosis rate (28.63%), with cells predominantly undergoing late apoptosis.

The present results suggested that glutamate might induce PC-12 cell apoptosis through activating the mitochondria-mediated apoptosis pathway. Caspase-3 is a key executor in this pathway and plays a critical role in the process of nuclear apoptosis, including DNA fragmentation and chromatin condensation [19,21]. In this experiment, caspase-3 activity was assessed based on the principle that caspase-3 catalyzes the substrate acetyl-Asp-Glu-Val-Asp p-nitroanilide, resulting in the production of yellow p-nitroanilide, which exhibits a strong absorption band around 405 nm. As shown in Figure 6B, glutamate incubation significantly promoted caspase-3 activity, whereas co-treatment with wushanicaritin or quercetin significantly reversed glutamate-induced enhancement. Similarly, gene expression levels of *caspase-3*, *caspase-7* and *Bax* were down-regulated by co-treatment with wushanicaritin, in comparison with the model group (Figure 6A). Additionally, a decreasing expression level of *Bcl-2* was caused by glutamate treatment. However, significant up-regulation was induced by wushanicaritin at 5 μM. 

Overproduction of ROS can lead to apoptotic cell death. And, mitochondria have been proven to be the important target of ROS [5,6]. Mitochondria-mediated apoptosis, regulated by Bcl-2 family members, triggers the permeabilization of the outer mitochondrial membrane, the release of cytochrome c and the activation of caspase-3 [40]. Both Bax and Bcl-2 belong to the Bcl-2 family and have a central role in regulating mitochondria-mediated apoptosis. Bax, a promoter of cell death, stimulates mitochondrial outer membrane permeabilization, leading to decreased Δψm and ultimately resulting in apoptosis, whereas Bcl-2 acts as an antiapoptotic factor that promotes cell survival by competitively binding to Bax. The ratio of Bax to Bcl-2 is a crucial indicator of cell apoptosis [41]. In the present work, a decrease in the ratio of *Bcl-2*/*Bax* was caused by glutamate incubation, while this trend was reversed in wushanicaritin-treated groups. Mitochondrial dysfunction, contributes significantly to the pathological processes of neurodegenerative diseases, and it is considered an important target for neuroprotective agents [4]. All of these results suggested that wushanicaritin might exert neuroprotective effects through inhibiting glutamate-induced mitochondria-mediated apoptosis.

## 4. Conclusions

In the current study, wushanicaritin displayed significant intercellular antioxidant activities and neuroprotective properties in PC-12 cells exposed to glutamate. The neuroprotective mechanisms of wushanicaritin were attributed to its role in suppressing ROS overproduction, protecting the enzymatic antioxidant defense system, maintaining mitochondrial function, and preventing cell apoptosis. As wushanicaritin is a characteristic bioactive compound found in *Epimedium*, the present results revealed its positive impact on the pharmacological effect of this medicinal plant, particularly its intercellular antioxidant and neuroprotective activities. Future research will delve deeper into the potential pharmacological properties of wushanicaritin, as well as the utilization of *Epimedium* species in neuroprotection.

## Figures and Tables

**Figure 1 foods-13-01493-f001:**
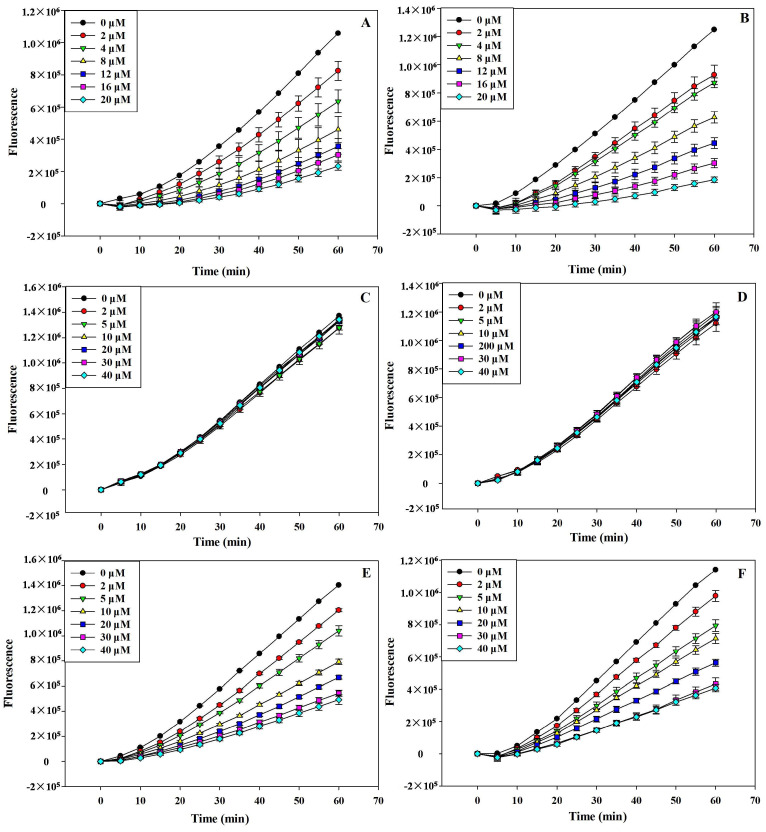
The peroxyl radical-induced oxidation of DCFH to DCF in HepG_2_ cells. The inhibition of DCFH oxidation by quercetin (**A**,**B**), incaritin (**C**,**D**), and wushanicaritin (**E**,**F**) over 1 h, using the protocol involving no PBS wash between antioxidant and ABAP treatments (**A**,**C**,**E**), and the protocol with a PBS wash (**B**,**D**,**F**) to remove antioxidants in the medium after incubation.

**Figure 2 foods-13-01493-f002:**
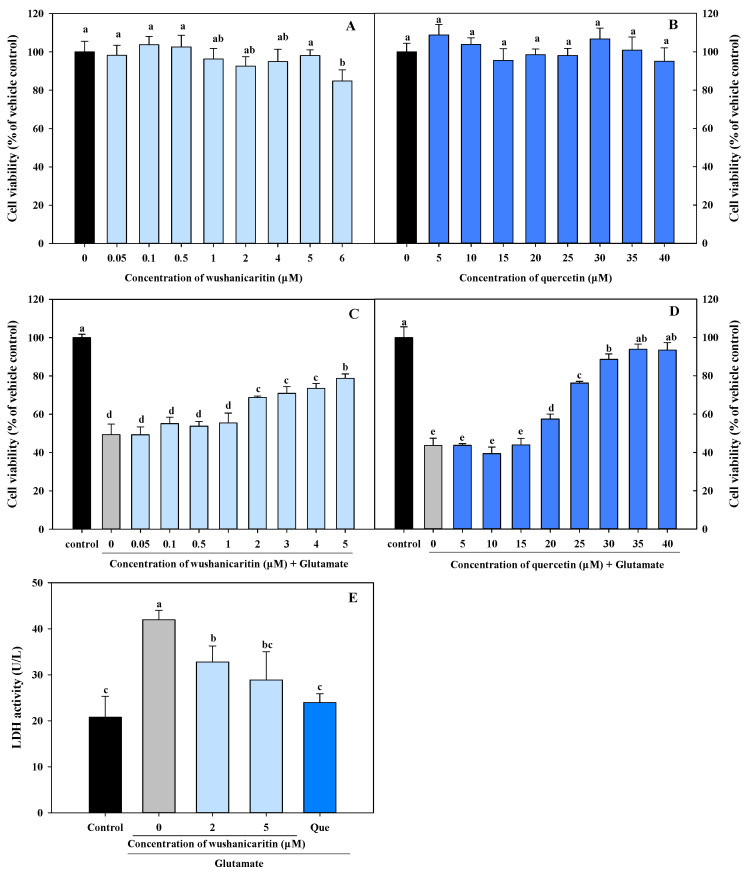
Effects of wushanicaritin and quercetin on the cell viability of PC-12 cells. (**A**) The viability of PC-12 cells incubated with wushanicaritin; (**B**) the viability of PC-12 cells incubated with quercetin; (**C**) the viability of PC-12 cells co-treated with glutamate and wushanicaritin; (**D**) the viability of PC-12 cells co-treated with glutamate and quercetin; (**E**) effects of wushanicaritin and quercetin on the LDH release. Que refers to quercetin. The bars with no letters in common exhibit significantly different values (*p* < 0.05).

**Figure 3 foods-13-01493-f003:**
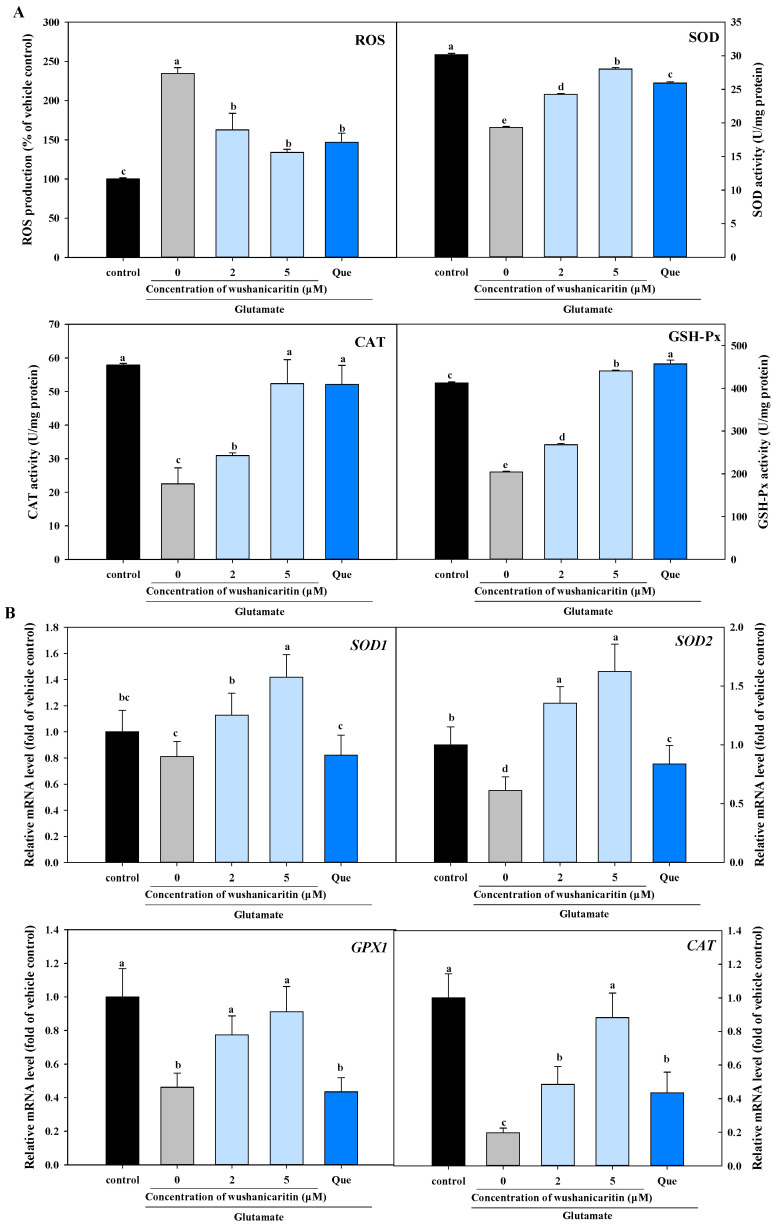
Effects of wushanicaritin and quercetin on the intracellular antioxidant defense system in glutamate-treated PC-12 cells. (**A**) Effects of wushanicaritin and quercetin on ROS production and intercellular antioxidant enzyme activities; (**B**) mRNA expression levels of genes related to oxidative stress. The mRNA expression levels were normalized to the vehicle control by using GAPDH as an endogenous reference. Que refers to quercetin. The values having no letters in common are significantly different (*p* < 0.05).

**Figure 4 foods-13-01493-f004:**
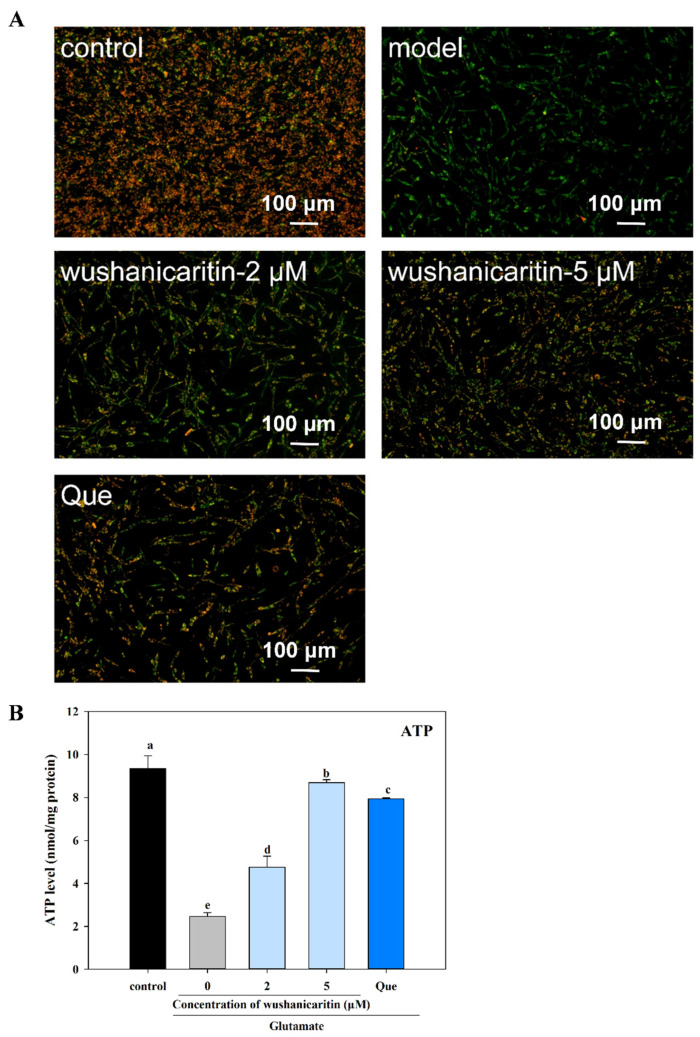
Effects of wushanicaritin and quercetin on the morphology of mitochondrial membrane potential (**A**) and ATP production (**B**). Control group refers to PC-12 cells with vehicle treatment (0.1% DMSO); Model group refers to PC-12 cells treated with glutamate; wushanicaritin-2 μM, wushanicaritin-5 μM and Que refers to PC-12 cells co-treated with glutamate and wushanicaritin at concentrations of 2, 5 μM, or quercetin (30 μM), respectively. Scale bar = 100 μm. The Bars with no letters in common are significantly different (*p* < 0.05).

**Figure 5 foods-13-01493-f005:**
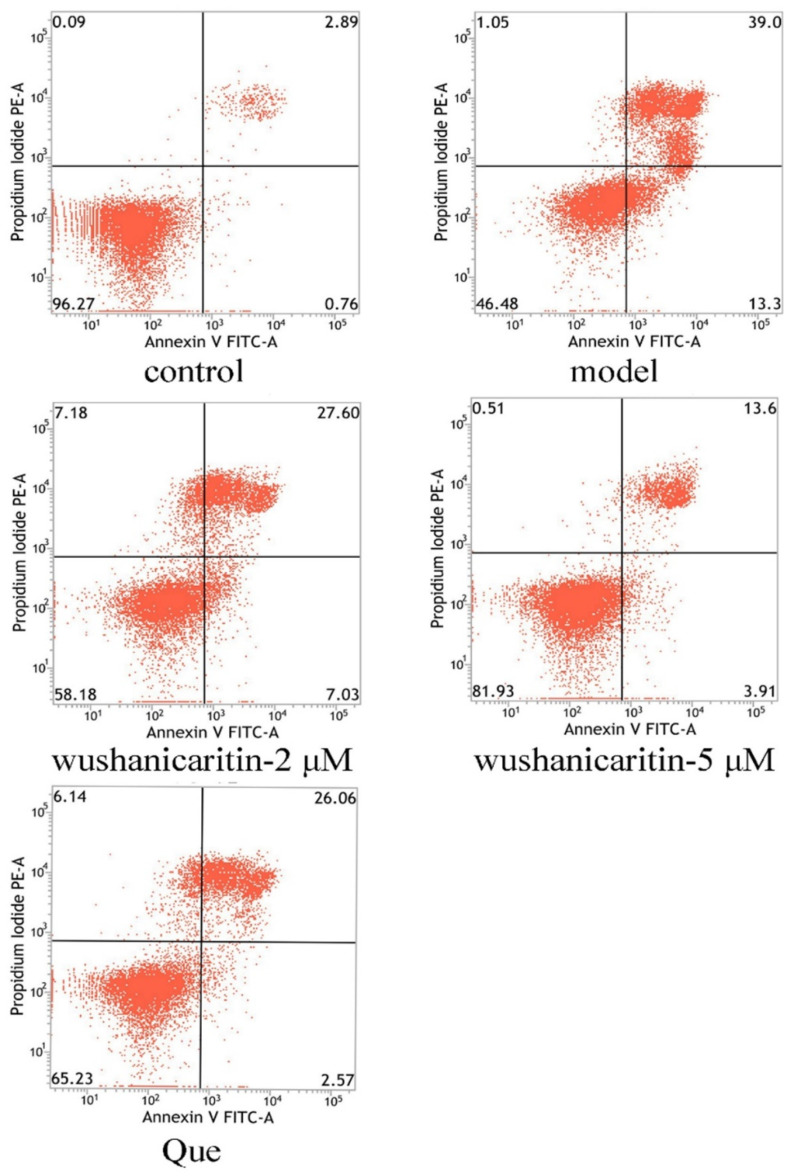
Effects of wushanicaritin and quercetin on cell apoptosis as determined by flow cytometry assay after Annexin V–FITC/PI staining. In the representative dot plots, Q1 indicates necrotic cells, Q2 indicates late apoptotic cells, Q3 indicates viable cells, and Q4 indicates early apoptotic cells. Control group refers to PC-12 cells with vehicle treatment (0.1% DMSO); Model group refers to PC-12 cells treated with glutamate; wushanicaritin-2 μM, wushanicaritin-5 μM and Que refer to PC-12 cells co-treated with glutamate and wushanicaritin at concentrations of 2, 5 μM, or quercetin (30 μM), respectively.

**Figure 6 foods-13-01493-f006:**
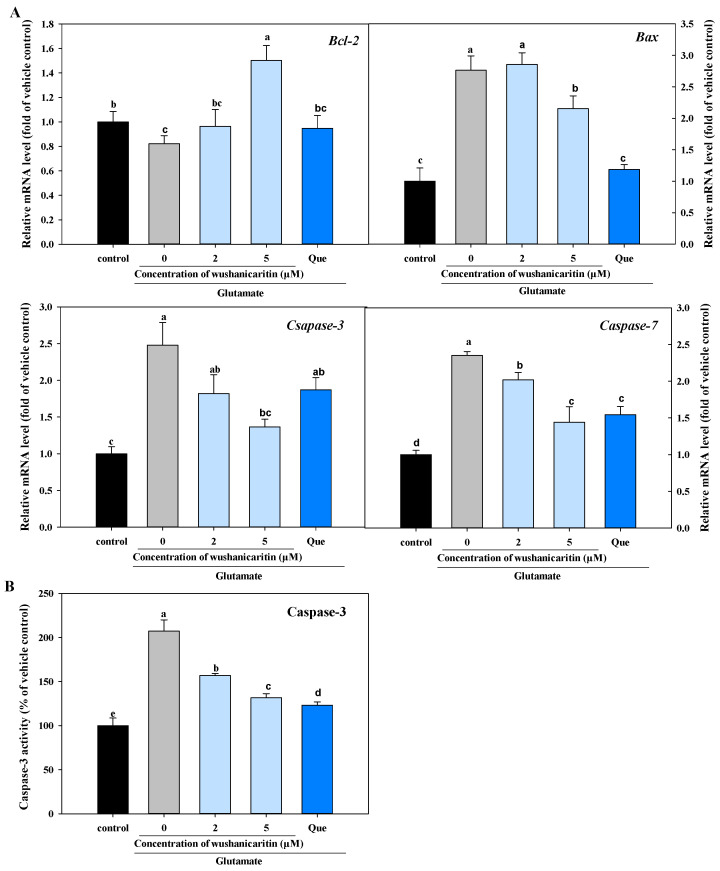
Effects of wushanicaritin and quercetin on the transcriptional levels of apoptotic-related genes (**A**) and caspase-3 activity (**B**). The mRNA expression levels were normalized to the vehicle control group by using GAPDH as an endogenous reference. Que refers to quercetin. The bars having no letters in common indicate significantly different values (*p* < 0.05) (Figure 1).

## Data Availability

The original contributions presented in the study are included in the article, further inquiries can be directed to the corresponding author.

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
