# Peer review of "From *Epimedium* to Neuroprotection: Exploring the Potential of Wushanicaritin"

_foods, 2024, doi:10.3390/foods13101493_

Round 1

Reviewer 1 Report

Comments and Suggestions for Authors

Dear Authors,

the paper is very interesting, very well written, with many techniques/methods, valuable results and potential for future research.

I have only few questions/sugesstions:

1.       3.2.2. section: Why didn't you determine the protein level of these genes, that would be the best evidence, because mRNA expression does not mean that the protein is produced?

2.       Figure 4. Is it possible to increase the red and green colors on the image to better see the increase or decrease in fluorescence?

3.       Conclusion: The conclusion is good, but you repeat the same line: "Wushanicaritin is a characteristic bioactive compound found in Epimedium species" (line 447 and line 452) - unnecessary repetition

Author Response

Response to the comments/recommendations from reviewer

We appreciate all the suggestions and comments from the reviewer. They are very helpful to improve our paper quality. The paper has been revised following these suggestions, and the revised words are marked in red color. A detailed response to all the reviewer suggestions is listed below. Please check. Hopefully, the revised manuscript can meet the requirements of Foods.

Reviewer #1:

Comment:

Dear Authors,

the paper is very interesting, very well written, with many techniques/methods, valuable results and potential for future research. I have only few questions/sugesstions.

Response:

Thank you for your positive comments. The paper has been revised as suggested.

Comment:

  1. 3.2.2. section: Why didn't you determine the protein level of these genes, that would be the best evidence, because mRNA expression does not mean that the protein is produced?

Response:

Thank you for the comments. This study has investigated the activities of intercellular antioxidant enzymes, including SOD, CAT, and GSH-Px. The results indicated that wushanicaritin and quercetin treatments could significantly (p< 0.05) increased the activities of these enzymes. Generally, an increase in enzyme activity corresponds to an increase in enzyme protein content, suggesting wushanicaritin treatment might promote these protein expression. Additionally, Alim et al. (2019) demonstrated that the transcriptional regulation of related genes, triggered by neuroprotectants, plays a pivotal role in preventing cell damage. Consequently, we determined the impact on the transcriptional levels of genes linked to antioxidant defense to further explore the mechanisms underlying wushanicaritin's neuroprotection. Nonetheless, understanding the protein levels of these genes is crucial for elucidating the mechanisms driving the neuroprotection induced by wushanicaritin. We plan to prioritize this aspect in our future research.

Relative reference:

Alim I.; Caulfield J.T.; Chen Y.; Swarup V.; Geschwind D.H.; Ivanova E.; Seravalli J.; Ai Y.; Sensing L.H.; Ste Marie E.J.; Hondal R.J.; Mukherjee S.; Cave J.W.; Sagdullaev B.T.; Karuppagounder S.S.; Ratan R.R. Selenium drives a transcriptional adaptive program to block ferroptosis and treat stroke. Cell 2019, 177, 1262-1279.

Comment:

  1. Figure 4. Is it possible to increase the red and green colors on the image to better see the increase or decrease in fluorescence?

Response:

Thank you for the suggestion. The quality of Figure 4 has been improved to facilitate a more detailed observation of the fluorescence variation.

Comment:

  1. Conclusion: The conclusion is good, but you repeat the same line: "Wushanicaritin is a characteristic bioactive compound found in Epimedium species" (line 447 and line 452) - unnecessary repetition

Response:

Thank you for the comment and suggestion. The sentence has been revised as “The current study demonstrated that wushanicaritin displayed significant inter-cellular antioxidant activities and neuroprotective properties in PC-12 cells exposed to glutamate.”.

Reviewer 2 Report

Comments and Suggestions for Authors

The authors here addressed an interesting topic on the potential of wushcaritin, phenylflavonoid from Epimedium species, as neuroprotoective agent. However, prior the publication some questions should be considered. 

Authors should explain more clearly how were the applied concentrations (dose ranges) chosen...
Also, should be discussed more on what would be potential application of the observed results. In other words, how this can be relevant to human health. Could this compound used in the formulation of novel foods/food supplements for neuroprotection.

Author Response

Response to the comments/recommendations from reviewer

We appreciate all the suggestions and comments from the reviewer. They are very helpful to improve our paper quality. The paper has been revised following these suggestions, and the revised words are marked in red color. A detailed response to all the reviewer suggestions is listed below. Please check. Hopefully, the revised manuscript can meet the requirements of Foods.

Reviewer #2:

Comment:

The authors here addressed an interesting topic on the potential of wushcaritin, phenylflavonoid from Epimedium species, as neuroprotoective agent. However, prior the publication some questions should be considered.

Response:

Thank you for the comments. The article has been revised as suggested.

Comment:

  1. Authors should explain more clearly how were the applied concentrations (dose ranges) chosen...

Response:

Thank you for the comments. The applied concentrations of tested samples were ascertained through preliminary experiments. It was meticulously ensured that these compounds exhibited no significant cytotoxicity to cells within the applied concentration ranges for both the cellular antioxidant activity assay and the cell neurotoxicity induction assay. For the assays aimed at understanding the underlying mechanisms of wushanicaritin's neuroprotective effect, functional concentration ranges were selected based on their significant enhancement of cell viability in the cell neurotoxicity induction assay. In this assay, wushanicaritin demonstrated a protective rate of 38.31% (the significant increase in cell viability at the lowest concentration among the tested doses) and 58.07% (the highest enhancement among the tested concentrations) against glutamate-induced damage at 2 and 5 μM respectively. Thereby, wushanicaritin at concentrations of 2 and 5 μM were chosen for further investigation. The relative information has been supplemented in the revised manuscript.

Comment:

  1. Also, should be discussed more on what would be potential application of the observed results. In other words, how this can be relevant to human health. Could this compound used in the formulation of novel foods/food supplements for neuroprotection.

Response:

Thank you for the comment and suggestion. In recent years, natural products, especially those with better antioxidant activities have been recognized as viable strategies for the prevention of neurodegenerative diseases and mitigation of associated side effects (Dilnashin et al., 2023; Osama et al., 2020). Furthermore, some bioactive compounds, notably those found in nutritional supplements and functional foods, have garnered significant attention (Khan et al., 2020). Epimedium has been widely utilized in Oriental countries, serving as a functional food, dietary supplements, or incorporated into wines, teas, tablets, extracts (Cui et al., 2023). As one of the important bioactive compound found in Epimedium, wushanicaritin showed promising intercellular antioxidant activities and neuroprotective effect, suggesting its potential application in functional foods or food supplements for neuroprotection. However, further research is required to substantiate these findings, including data from animal studies and clinical trials. The relative information has been supplemented in the revised manuscript.

Relative references:

Cui, Q., Wang, C., Zhou, L., Wei, Y., Liu, Z., Wu, X., 2023. Simple and novel icariin-loaded pro-glycymicelles as a functional food: physicochemical characteristics, in vitro biological activities, and in vivo experimental hyperlipidemia prevention evaluations. Food Funct. 14(21), 9907-9919.Dilnashin, H., Birla, H., Keswani, C., Singh, S. S., Zahra, W., Rathore, A. S., Singh, R., Keshri, P. K., Singh, S. P., 2023. Neuroprotective effects of Tinospora cordifolia via reducing the oxidative stress and mitochondrial dysfunction against rotenone-induced PD mice. ACS Chem. Neurosci. 14(17), 3077-3087.

Khan, H., Tundis, R., Ullah, H., Aschner, M., Belwal, T., Mirzaei, H., Akkol, E. K., 2020. Flavonoids targeting NRF2 in neurodegenerative disorders. Food Chem. Toxicol. 146, 111817.

Osama, A., Zhang, J., Yao, J., Yao, X., Fang, J., 2020. Nrf2: a dark horse in Alzheimer's disease treatment. Ageing Research Reviews 64, 101206.
